# Inactive matrix gla protein plasma levels are associated with peripheral neuropathy in Type 2 diabetes

**Anne-Caroline Jeannin**[1,2], **Joe-Elie Salem**[1,3,4,5], **Ziad Massy**[6], **Carole Elodie Aubert**[7,8], **Cees Vemeer**[9], **Chloé Amouyal**[1,2,5], **Franck Phan**[1,2,5,10], **Marine Halbron**[1,2,5], **Christian Funck-Brentano**[1,3,4,5], **Agnès Hartemann**[1,2,5,10], **Olivier Bourron**[1,2,5,10]*

**1** Sorbonne Université, Paris, France, **2** Assistance Publique-Hôpitaux de Paris (APHP), Diabetology Department, Pitié-Salpêtrière Hospital, Paris, France, **3** Department of Pharmacology and CIC-1421, AP-HP, Pitié-Salpêtrière Hospital, Paris, France, **4** INSERM, CIC-1421, Paris, France, **5** Institute of Cardiometabolism and Nutrition ICAN, Paris, France, **6** Division of Nephrology, Ambroise Paré Hospital, AP-HP, Pitié-Salpêtrière Hospital, Université Paris-Saclay, Paris, France, **7** Department of General Internal Medicine, Inselspital, Bern University Hospital, University of Bern, Bern, Switzerland, **8** Institute of Primary Health Care (BIHAM), University of Bern, Bern, Switzerland, **9** Cardiovascular Research Institute CARIM, Maastricht University, Maastricht, The Netherlands, **10** INSERM, UMR_S 1138, Centre de Recherche des Cordeliers, Paris, France

* olivier.bourron@aphp.fr

**Data Availability Statement:** All data from the current study were reported in the manuscript and tables. However, there was not an authorization for making data set publicly available when the study

## Abstract

### Aims/Hypothesis

Diabetic peripheral neuropathy is a frequent and severe complication of diabetes. As Matrix-gla-protein (MGP) is expressed in several components of the nervous system and is involved in some neurological disease, MGP could play a role in peripheral nervous system homeostasis. The aim of this study was to evaluate factors associated with sensitive diabetic neuropathy in Type 2 Diabetes, and, in particular, dephospho-uncarboxylated MGP (dp-ucMGP), the inactive form of MGP.

### Methods

198 patients with Type 2 Diabetes were included. Presence of sensitive diabetic neuropathy was defined by a neuropathy disability score (NDS) $\geq$6. Plasma levels of dp-ucMGP were measured by ELISA.

### Results

In this cohort, the mean age was 64+/-8.4 years old, and 80% of patients were men. Peripheral neuropathy was present in 15.7% of the patients and was significantly associated (r = 0.51, p<0.0001) with dp-ucMGP levels ($\beta$ = -0.26, p = 0.045) after integrating effects of height ($\beta$ = -0.38, p = 0.01), insulin treatment ($\beta$ = 0.42, p = 0.002), retinopathy treated by laser ($\beta$ = 0.26, p = 0.02), and total cholesterol levels ($\beta$ = 0.3, p = 0.03) by multivariable analysis.

protocol was submitted for the Local Ethical Committee. Therefore, data are available upon request to Dr Alban Danset at: "alban. danset@aphp.fr".

**Funding:** This work was supported by a fund from the Lilly Company. The research activities of C.E.A. were supported by a doctoral research scholarship from the University of Lausanne. A.C.J received a grant from Ministère français des Affaires Sociales et de la Santé (Bourse année recherche 2017/ 2018) The company was involved neither in the design of the study nor in data collection.

**Competing interests:** A patent has been filed on a method using circulating Matrix Gla protein measurement for diagnosis and treating peripheral neuropathies by Assistance Publique Hôpitaux de Paris - APHP). Olivier Bourron, Joe-Elie Salem and Agnès Hartemann are the inventors. The application number is 18306503.6 – 1118. Status of application: recorded. The patent specific aspect of manuscript covered in patent application: use of circulating Matrix Gla protein measurement for diagnosis and treating peripheral neuropathies. This study was supported by a fund from Lilly Company. The company was involved neither in the design of the study nor in data collection. This does not alter our adherence to PLOS ONE policies on sharing data and materials.

## Conclusions

The association between diabetic neuropathy and the inactive form of MGP suggests the existence of new pathophysiological pathways to explore. Further studies are needed to determine if dp-ucMGP may be used as a biomarker of sensitive neuropathy. Since dp-ucMGP is a marker of poor vitamin K status, clinical studies are warranted to explore the potential protective effect of high vitamin K intake on diabetic peripheral neuropathy.

## Introduction

Diabetic peripheral neuropathy is a frequent complication of diabetes. It affects about 10 to 15% of patients with Type 2 Diabetes at diagnosis and up to 50% after 10 years of disease duration [1]. Diabetic neuropathy is associated with high morbidity and mortality [2], because of increased risk for foot ulceration and amputation [3], and for poor quality of life and depression [4]. So, it is related to high healthcare costs [5]. The main clinical characteristic of diabetic peripheral neuropathy is a decrease of distal sensitivity that represents the most important risk factor of foot ulceration in patients with diabetes. In 2019, ADA guidelines recommended an annual clinical screening to diagnose sensitive diabetic neuropathy [1]. ADA recommendations for screening and diagnosis include a careful history and assessment of either temperature or pinprick sensation (small-fiber function) and vibration sensation using a 128-Hz tuning fork (for large-fiber function). All patients should have annual 10-g monofilament testing to identify feet at risk for ulceration and amputation. Electrophysiological testing or referral to a neurologist is not recommended for screening, except in situations where the clinical features are atypical, the diagnosis is unclear, or a different etiology is suspected [1].

Mechanisms involved in diabetic neuropathy are not clearly understood. The main hypothesis is that chronic glucotoxicity and lipotoxicity lead to oxidative stress, inflammation, and mitochondrial dysfunction and finally to nerve damage with neuron degeneration and demyelination [6, 7].

Matrix gla protein (MGP) is an 84 amino acids protein containing five glutamic acid residues (glu residues) and three serine residues. MGP exists in inactive and active forms [8]. The activation of MGP is obtained after a vitamin K dependent gamma-glutamyl carboxylation of glutamic acid residues (forming gla residues) and a phosphorylation of serine residues [9, 10]. Desphospho-uncarboxylated MGP (dp-ucMGP) represents therefore the inactive form of MGP. MGP was initially isolated from bone tissue but it is also expressed by chondrocytes, vascular smooth muscle cells, endothelial cells but also neurons and glial cells [11, 12]. Moreover, several studies suggest that MGP plays a role in the nervous system. First, in 2005, a novel mutation of MGP associated with high level of inactive dp-ucMGP is described and associated with neurological manifestations, abnormalities of brain's white matter and optic nerve atrophy, in addition to typical manifestations of Keutel syndrome [13, 14], suggesting a link between MGP activity and nervous system pathophysiology. Then, Goritz et al have demonstrated that MGP is expressed by neurons, and is regulated by glial cells [12]. Finally, some studies reveal also that MGP could be implicated in neurological disease, as glioblastoma [15], and Alzheimer disease [16].

Given that the pathogenesis of diabetic neuropathy remains unclear and that MGP could be involved in nervous system pathophysiology, we hypothesize that MGP may be involved in diabetic peripheral neuropathy development. The objective of this study is to evaluate the clinical and biological markers, in particular inactive dp-ucMGP, associated with diabetic peripheral neuropathy on patients with Type 2 Diabetes.

## Material and methods

### Study design

This study is a cross-sectional ancillary study to the prospective DIACART cohort[17]. DIA-CART cohort was initially designed to study the clinical and biological variables associated with peripheral arterial calcification and other diabetic complications such as diabetic neuropathy. In this cohort, diabetic peripheral neuropathy was accurately assessed with a careful foot examination to calculate the NDS score [7]. 198 Patients were recruited in the diabetes and cardiology departments, in the Pitié-Salpêtrière hospital (APHP, Paris, France), over eight months, from November 2011 to July 2012. They were subsequently prospectively assessed in a cardio-metabolic clinical research center (INSERM, CIC 1421) for clinical phenotyping and bio banking of their blood samples.

### Participants

DIACART cohort was initially designed to study the clinical and biological variables associated with diabetic complications [17]. The study focused on patients with Type 2 Diabetes, at high cardiovascular risk. Inclusion criteria were Type 2 Diabetes with at least one of the following criteria: coronary artery disease or peripheral arterial occlusive disease or age>50 years for men or >60 years for women. Exclusion criteria were an estimated glomerular filtration rate calculated with the modification of diet in renal disease <30ml/min and a history of lower limb angioplasty and/or bypass. The peripheral nerve deficit of nondiabetic origin (e.g. alcohol, neurotoxic medications (e.g., chemotherapy), vitamin B12 deficiency, hypothyroidism, renal disease, malignancies (e.g., multiple myeloma), infections (e.g., HIV, HCV), chronic inflammatory demyelinating neuropathy, inherited neuropathies, compression due to vertebral disk herniation, and vasculitis) was excluded through a careful medical history review, a differential test or both.

### Informed consent and ethical aspect

The study was approved by the local ethics committee (PARIS VI CPP) and registered in ClinicalTrials.gov (Identifier: NCT02431234). All patients were informed on the study objectives and procedure. Participants gave their written informed consent to participation. All methods were carried out in accordance with relevant guidelines and regulations.

### Procedure

Data collection, including a clinical evaluation and blood tests, were realized during a one-day hospitalization in a cardio-metabolic clinical research center.

### Diabetic peripheral neuropathy

In this cohort, diabetic peripheral neuropathy was accurately assessed with a careful foot examination, including several physical tests [7]. Diabetic peripheral neuropathy was assessed by the modified neuropathy disability score (NDS), scoring from 0 to 10 [18]. NDS assesses vibration sensory on the great toe using 128-Hz tuning fork, temperature sensory on dorsum of the foot using tubes of ice or warm water, pinprick sensory applying pin near to big toe nail and Achilles reflex. Each sensory test scores 0 for normal and 1 for abnormal sensation, for each foot. Achilles reflex score 0 if they are present, 1 if they are present with reinforcement and 2 if they are absent, for each foot. NDS $\geq$ 6 allows the diagnosis of diabetic peripheral neuropathy [19]. The NDS was also used as a continuous variable to assess magnitude of peripheral neuropathy because NDS score is a validated and widely used score for detecting neuropathy.

## Clinical data

During the patient interview, the physician collected medical information about personal disease history, comorbidities and treatment. Clinical tests were conducted by a physician blinded to blood tests results.

## Biochemical measures

Blood and urine samples were collected in the morning fasting for the measurement of biochemistry analyses including hemoglobin A1c (HbA1c), high-sensitivity C-reactive protein (hsCRP), estimated glomerular filtration rate (eGFR) by modification of diet in renal disease (MDRD), urinary albumin/creatinine ratio, serum calcium corrected for albumin, serum phosphorus, total cholesterol, triglycerides and IL-6.

Assays were developed to measure dp-ucMGP in plasma [14]. These assays were conducted after the samples freezing, storage at -80°C and thawing. Dp-ucMGP levels were measured by a dual-antibody ELISA. The capture antibody was directing against the non-phosphorylated MGP sequence 3–15 (mAb-dpMGP; VitaK BV, Maastricht, The Netherlands) and the detecting antibody was directed against the uncarboxylated MGP sequence 35–49 (mAb-ucMGP; VitaK BV). Intra-assay variability was 5.6% for dp-ucMGP. Inter-assay variability was 9.9% for dp-ucMGP. Dp-ucMGP was measured in archived samples of 81 age-matched controls. The mean levels were respectively 557+/-277 pmol/l (median: 522 pmol/l).

## Statistical analyses

Data were described as mean +/- standard deviation of the mean or frequency, as appropriate. Comparison of quantitative variables was performed using Student's t test or Mann-Whitney test, when variables were normally and non-normally distributed, respectively. Comparison of qualitative variables was performed using $\chi2$ test. Pearson's coefficient (r) was used to assess association between quantitative variables. A 95% confidence interval for the correlation coefficient was calculated using Fisher's method (Prism 6; GraphPad Software, Inc). Multivariable analyses were performed by ANCOVA (continuous NDS scoring) or logistic regression (Presence/absence of neuropathy defined by NDS≥6/<6). Only covariates with significant univariate association (In bold, in Tables 1 and 2) with NDS were further integrated for multivariate analyses (XLstat-software, Addinsoft®, New-York). For multivariate analysis, beta-coefficients (β) were calculated to allow for direct comparison of the relative influence of the explanatory variables on the dependent variable, and their significance (P≤0.05 considered significant).

In this cohort (n: 198), the study had a power≥80% to detect a significant correlation (with r≥0.2, α-risk: 0.05, Student approximation) between each clinical or biological variable and NDS score.

## Results

### Baseline characteristics

Clinical and biological characteristics at baseline for the total cohort, and for patients with and without neuropathy are described in Table 1. Finally, 198 patients were included in the DIA-CART study, 80% of whom were men. Study participants were young-old (64+/-8.4 years old) and overweight (mean BMI of 29,16±5.3 kg/m$^2$) patients. Their mean height was 1.7+/-0.08 meters. Diabetes duration was 14.6+/-9.3 years, and mean HbA1c was 7.8%+/-1.5% (61.8 +/-16.2 mmol/L). Concerning diabetes comorbidities, 14.1% of patients had a retinopathy treated with laser, 36% had a urinary albumin/creatinine ratio >3 mg/mmol, and mean eGFR

**Table 1. Baseline characteristics of the patients.**

| Characteristics | Total cohort | Neuropathy (NDS≥6) | Without neuropathy (NDS<6) | p-value |
|---|---|---|---|---|
| N (%) | 198 | 31 (15.7) | 167 (84.3) | - |
| Age, years | 64±8.4 | 64±8.6 | 64±8.4 | ns |
| Male, n(%) | 158 (79.8) | 26 (83.9) | 132(79) | ns |
| **Height (cm)** | 170±8 | 173±7 | 169±8 | **0.009** |
| BMI (Kg/m2) | 29,16±5.3 | 30,23±5.5 | 28,97±5.2 | ns |
| Diabetes duration, years | 14.6±9.3 | 14.6±10.2 | 14.6±9.2 | ns |
| Hypertension, n (%) | 163 (82.3) | 28 (90.3) | 135 (80.8) | ns |
| **NDS score, points** | 2.4±2.4 | 6.8±1.5 | 1.6±1.5 | **<0.0001** |
| **Retinopathy treated with laser, n(%)** | 28 (14.1) | 10(32.3) | 19(11.3) | **0.003** |
| **Coronary arterial disease, n(%)** | 150 (75.8) | 28 (90.3) | 122 (73.05) | **0.04** |
| Ischemic stroke, n(%) | 14 (7.1) | 2(6.5) | 12(7.2) | ns |
| eGFR calculated by MDRD, mL/min | 76±20 | 72±20 | 77±20 | ns |
| **Urinary albumin /creatinine ratio>3 (mg/mmol), n(%)** | 71 (35.9) | 19(61.3) | 52 (31.1) | **0.001** |
| **Insulin treatment, n(%)** | 94 (47.5) | 24(77.4) | 70(41.9) | **0.0003** |
| HbA1c, mmol/mol | 61.8±16.2 | 66.6±20.5 | 60.9±15.2 | ns |
| HbA1c, % | 7.8±1.5 | 8,2±1.9 | 7.7±1.4 | ns |
| hsCRP, mg/L | 2.2±2.5 | 2.4±2.8 | 2.2±2.5 | ns |
| IL-6, pg/mL | 5±22 | 4.6±3.4 | 5.3±24 | ns |
| Corrected calcium, mmol/L | 2.3±0.1 | 2.3±0.3 | 2.3±0.1 | ns |
| Phosphorus, mmol/L | 1.02±0.15 | 1.02±0.14 | 1.02±0.16 | ns |
| Triglycerides, mmol/L | 1.6±1.1 | 1.5±0.8 | 1.6±1.1 | ns |
| **Total cholesterol, mmol/L** | 3.7±0.9 | 3.4±0.8 | 3.8±0.9 | **0.02** |
| **dp-ucMGP, pmol/L** | 627±451 | 821±703 | 591±379 | **0.009** |

Quantitative variables are represented by mean ± standard deviation. Data are given as the number (percentage) for binary variables. Data are no significant (ns) if p>0.05. Significant differences between patients with and without neuropathy are in bold.

Abbreviations: BMI body mass index, eGFR MDRD estimated glomerular filtration rate calculated with the modification of diet in renal disease formula, HbA1c haemoglobin A1C, hsCRP high sensitivity C-reactive protein, IL-6 interleukin 6, dp-ucMGP dephospho-uncarboxylated matrix gla protein, NDS neuropathy disability score.

calculated by MDRD was 76+/-20 ml/min. Mean NDS was 2.4+/-2.4 points, and 15.7% of subjects had a diabetic peripheral neuropathy, defined by NDS≥6. The mean level of dp-uc MGP was 627 +/-451 pmol/l.

## Factors associated with diabetic neuropathy (defined by NDS≥6)

Patients with neuropathy were significantly taller (173 cm vs 169 cm, p = 0.009) than patients without neuropathy. Cholesterol total was significantly lower in patients with neuropathy compared to patients without neuropathy (3.8 mmol/L vs 3.4 mmol/L, p = 0.02). Retinopathy treated with laser (32 vs 11%, p = 0.003), urinary albumin/creatinine ratio >3 mg/mmol (61 vs 31%, p = 0.001), coronary arterial disease (90 vs 73%, p = 0.04) and insulin treatment (77 vs 42%, p = 0.0003) were significantly more common in patients with neuropathy. Age, sex ratio, diabetes duration and HbA1c were not different between patients with and without neuropathy (Table 1). Dp-ucMGP levels were significantly higher in patients with neuropathy than in those without neuropathy (821 vs 591 pmol/l respectively, p = 0.009). In multivariate analysis integrating all significant covariates (retinopathy treated with laser, urinary albumin/creatinine ratio, coronary arterial disease, insulin treatment and quantitative variables in bold, in Tables 1 and 2), presence of neuropathy defined by NDS score ≥6 was still associated

**Table 2. Univariate analysis: Correlations between clinical and biological variables and NDS.**

| | r, [CI 95%] | p-value |
|---|---|---|
| Age | 0.07 [-0.08; 0.20] | ns |
| **Height** | **0.25 [0.11; 0.38]** | **0.0004** |
| **Body mass index (kg/m$^2$)** | **0.09 [-0.05; 0.23]** | ns |
| Diabetes duration | 0.03 [-0.11; 0.17] | ns |
| **eGFR** | **-0.16 [-0.29; -0.02]** | **0.03** |
| **HbA1c** | **0.21 [0.08; 0.34]** | **0.04** |
| HsCRP, mg/L | 0.03 [-0.11; 0.17] | ns |
| IL-6, pg/mL | 0.08 [-0.07; 0.21] | ns |
| Corrected calcium, mmol/L | 0.08 [-0.06; 0.22] | ns |
| Phosphorus, mmol/L | 0 [-0.14; 0.14] | ns |
| Triglycerides | -0.07 [-0.21; 0.07] | ns |
| Total cholesterol | -0.11 [-0.25; 0.02] | ns |
| **dp-ucMGP** | **0.22 [0.08; 0.34]** | **0.002** |

Correlations were performed by Pearson's coefficient (r). 95% confidence interval of the correlation coefficient was assessed using Fisher's method, and is presented in brackets. Correlations are significant if p<0.05. Significant results are presented in bold.

Abbreviations: BMI body mass index, eGFR MDRD estimated glomerular filtration rate calculated with the modification of diet in renal disease formula, HbA1c haemoglobin A1C, hsCRP high sensitivity C-reactive protein, IL-6 interleukin 6, dp-ucMGP dephospho-uncarboxylated matrix gla protein, NDS neuropathy disability score.

(r = 0.51, p<0.0001) with dp-ucMGP levels (β = -0.26, p = 0.045), height (β = -0.38, p = 0.01), insulin treatment (β = 0.42, p = 0.002), retinopathy treated by laser (β = 0.26, p = 0.02), and total cholesterol level (β = 0.3, p = 0.03) (Table 3).

## Factors associated with continuous NDS scoring

In univariate analysis (Table 2), NDS was positively associated with height (r = 0.25, p = 0.0004), HbA1c (r = 0.21, p = 0.04) and dp-ucMGP (r = 0.22, p = 0.002). NDS was negatively associated with eGFR (r = -0.16, p = 0.03). In multivariate analysis integrating all significant covariates (in bold, in Tables 1 and 2), NDS scoring was still associated (r = 0.51, p<0.0001) with dp-ucMGP levels (β = 0.16, p = 0.025), height (β = 0.29, p<0.0001), HbA1c (β = 0.19, p = 0.006), insulin treatment (β = 0.19, p = 0.007), retinopathy treated by laser (β = 0.16, p = 0.015) and urinary albumin/creatinine ratio>3 mg/mmol (β = 0.14, p = 0.031) (Table 4).

**Table 3. Multivariate analysis: Correlations between clinical and biological variables and diabetic neuropathy (NDS ≥ 6).**

| | β, [95% confidence interval] | p-value |
|---|---|---|
| **Height** | **-0.38, [-0.67–0.09]** | **0.01** |
| **Retinopathy treated with laser** | **0.26, [0.05–0.47]** | **0.02** |
| **Insulin treatment** | **0.42, [0.15–0.7]** | **0.002** |
| **Total cholesterol** | **0.3, [0.03–0.57]** | **0.03** |
| **dp-ucMGP** | **-0.26, [-0.51–0.01]** | **0.045** |

Multivariate analysis was performed using ANCOVA. 95% confidence interval of the standardized coefficient is presented in brackets. Correlations are significant if p<0.05. Significant results are presented in bold.

Abbreviations: β: standardized coefficient, dp-ucMGP dephospho-uncarboxylated matrix gla protein, NDS neuropathy disability score.

**Table 4. Multivariate analysis: Correlations between clinical and biological variables and continuous NDS scoring.**

| | β, [95% confidence interval] | p-value |
|---|---|---|
| **Height** | **0.29, [0.16–0.41]** | **<0.0001** |
| **Retinopathy treated with laser** | **0.16, [0.03–0.29]** | **0.015** |
| **Insulin treatment** | **0.19, [0.05–0.33]** | **0.007** |
| **Urinary albumin/creatinine ratio>3** | **0.14, [0.01–0.28]** | **0.031** |
| **HbA1c** | **0.19, [0.06–0.33]** | **0.006** |
| **dp-ucMGP** | **0.16, [0.02–0.29]** | **0.025** |

Multivariate analysis was performed using ANCOVA. 95% confidence interval of the standardized coefficient is presented in brackets. Correlations are significant if p<0.05. Significant results are presented in bold. Abbreviations: β: standardized coefficient, dp-ucMGP dephospho-uncarboxylated matrix gla protein, HbA1c haemoglobin A1C, NDS neuropathy disability score.

## Discussion

This study reveals that peripheral neuropathy, defined by a NDS score≥6, in type 2 diabetic patients is significantly associated with height, insulin treatment, retinopathy treated with laser, total cholesterol and, particularly to dp-ucMGP plasma levels. These factors, HbA1c and urinary albumin/creatinine ratio>3 mg/mmol are also associated with the magnitude of NDS scoring.

Height, poor glycemic control and dyslipidemia are known risk factors of diabetic neuropathy [4]. We don't find any significant association between BMI and neuropathy in our study, probably because BMI formula includes height, which is maybe a more important marker of diabetic peripheral neuropathy due to the length-dependent presentation of this neuropathy [20]. Furthermore, BMI is not also associated with neuropathy in the DIACART study probably because majority of the patients included were overweight or obese (mean BMI 29.16 +/-5.3 Kg/m$^2$). Although insulin is considered as a neurotrophic factor and although low-dose insulin can have beneficial effects on diabetic neuropathy, insulin use is associated with diabetic neuropathy in the DIACART study [21]. Retinopathy and nephropathy are usual comorbidities of diabetic neuropathy, explaining their association in this study [4]. In the same way, age and gender are not related to neuropathy in our population because most of the patients included in this study were male (80%) and elderly (mean age 64+/-8.4 years old).

The most important and original result is the association between dp-ucMGP plasma levels and diabetic neuropathy. Moreover, dp-ucMGP plasma levels increased with the continuous NDS scoring.

This association could be explained by several hypotheses.

We can suppose that the association between dp-ucMGP and neuropathy could directly result from MGP involvement in pathophysiology of diabetic peripheral neuropathy. Indeed, MGP is a protein from extracellular matrix mainly expressed in osteoarticular and vascular systems, but Goritz et al have shown that MGP is also expressed by neurons and glial cells [12]. The main ligand of MGP is Bone Morphogenetic Protein-2 (BMP-2) [22] and some data show that MGP, via modulation of BMP-2 signaling, could participate in the early differentiation and growth of neurons, in dendrites formation, in the development of mature Schwann cells and in the myelination [23–26]. Furthermore, MGP can also interact with fibronectin, which is involved in axon regeneration by its interaction with Schwann cells [27–29]. Consequently, an excess of inactive form of MGP (i.e., dp-ucMGP), could be associated with nerve damage and a source of axon regeneration loss, two pathological conditions observed in diabetic neuropathy.

However, dp-ucMGP is primarily an inverse marker for vitamin K status, and the association between dp-ucMGP and neuropathy suggests that poor vitamin K status is an independent risk factor for diabetic peripheral neuropathy. Comparison of different MGP assays showed that the dp-ucMGP assay is particularly suited to assess vascular vitamin K status and dp-ucMGP is then considered as the most sensitive biomarker for poor vitamin K status presently known [30]. Poor vitamin K status, estimated by dp-ucMGP, has been before described as associated with increased cardiovascular risk in Type 2 Diabetes [31]. Here we show, for the first time, an association between vitamin K status and diabetic peripheral neuropathy. The role of vitamin K in the nervous system was initially described via observations of microcephaly, optic atrophy and mental retardation resulting from fetal exposure to warfarin [32]. A recent study has shown that vitamin K enhances, during remyelination, the production of brain sulfatides, the sulfated form of galactosylceramides [33]. Decreases in myelin sulfatides content have been implicated as important factors in the disruption of myelin stability and function [34]. Furthermore, vitamin K seems to have survival-promoting effect on neurons [35]. We can therefore hypothesize that low vitamin K status could be associated with myelin alteration and cytotoxic effects on neurons in peripheral nerve tissue. Further studies are needed to confirm the role of vitamin K in peripheral diabetic neuropathy.

Vitamin K is an essential cofactor for the maturation of several proteins, not only for MGP. We cannot therefore exclude that the association of poor vitamin K status with diabetic peripheral neuropathy may also be a marker of the involvement of another vitamin K-dependent protein that is important for the neural system.

Circulating inactive dp-ucMGP could also be useful as a biomarker of diabetic neuropathy. The diagnosis of diabetic neuropathy is mainly clinical, based on sensory tests. But, these tests need to be associated to increase their sensitivity, are operator-dependent and time-consuming. Different surveys revealed that about only 65% of patients with diabetes yearly had a foot examination by a physician [36]. So, biomarker of diabetic neuropathy could be useful for clinical practice. Several biomarkers have been suggested, as neuron-specific enolase, toll-like receptor 4 or tumor necrosis factor alpha (TNF-α), but they are not specific of diabetic neuropathy [37, 38]. Further studies are needed to clarify if dp-ucMGP could be a good biomarker in this field.

There is some data showing that dp-ucMGP is associated with several micro and macrovascular complications of diabetes, including diabetic nephropathy, retinopathy, vascular stiffness and vascular calcification [39–41]. Although dp-ucMGP has been repeatedly associated with vascular calcification and cardiovascular disease, dp-ucMGP is not associated in our study with coronary arterial disease (S1 Table) [39, 41]. As observed by others, dp-ucMGP is associated negatively with eGFR estimated by MDRD and with albuminuria (S1 Table) [39, 40]. However we don't find any association between dp-ucMGP and retinopathy treated by laser (S1 Table) despite some data suggesting that dp-ucMGP could be a marker of retinal health [42]. Additional studies are needed to explore specifically these associations in patients with diabetes.

Since dp-ucMGP is a vitamin K dependent protein, diabetic peripheral neuropathy in our study is associated with poor vitamin K status. Clinically, this gives possibilities to explore options for treatment with vitamin K supplements and especially to put in place preventive measures in diabetic patients at risk of peripheral neuropathy. The treatment of diabetic neuropathy remains currently mainly symptomatic, based on pain treatment. Targeted therapies have been developed: aldose reductase inhibitors, blocking the polyol pathway, protein kinase C inhibitors, and aminoguanidine, preventing the synthesis of age glycation end products. Despite promising results in pre-clinical animal models, clinical trials haven't demonstrated any benefit in man [43–46]. Vitamin K supplementation is safe in human and interventional

studies are needed to determine if vitamin K supplementation could prevent diabetic peripheral neuropathy [47, 48].

So, further studies are really warranted to better understand the role of MGP or other vitamin K dependent protein in the peripheral nervous system. These studies could be led in larger cohorts of patients with Type 2 Diabetes patients, and in cohorts of patients with Type 1 Diabetes or with other neuropathies, in order to analyze if this association is specific to diabetic peripheral neuropathy or not. Depending on the results of these studies, MGP could be used in diabetic neuropathy for diagnosis, prediction or therapeutic purposes via vitamin K supplementation.

The strengths of this study are the accurate diagnosis of diabetic neuropathy by a validated score (NDS), and by the same physician for all patients. Although electrophysiological tests and pathological tests, that are gold standards for the evaluation of diabetic polyneuropathy, were not performed in the DIACART study, we used NDS score, a largely validated test for neuropathy diagnosis [18, 49–56]. NDS is well correlated with neurophysiological and sural nerve morphometric abnormalities in patients with diabetes [49, 51, 57–59]. So the NDS is a widely used and widely accepted scoring test for diabetic neuropathy. Moreover, in the DIACART study, most of the common risk factors for neuropathy were associated to NDS score. The subject of this study is innovative. Indeed, this is the first study to concomitant evaluate dp-ucMGP (i.e. low vitamin K status) and peripheral neuropathy in patients with Type 2 Diabetes and we describe here, for the first time, their association. The study would have been strengthened by the presence of a control group of non-diabetic participants, so that they could have been compared with diabetic patients. Other limitations of this study are the small number of patients with a neuropathy, the cross-sectional design which allows only association but not causal relationships and the absence of the gold standard test, sural nerve biopsy, to define diabetic neuropathy.

To conclude, this study suggests that that dp-ucMGP and poor vitamin K status are associated with peripheral diabetic neuropathy in Type 2 Diabetes and that dp-ucMGP could be a biomarker of choice to identify subjects at risk of diabetic neuropathy. Further studies are warranted to precise if circulating MGP is a biomarker and/or a causal factor of diabetic neuropathy. Then fundamental experiments and prospective clinical studies are needed to clarify the role of MGP in diabetic neuropathy, and if this protein could be used as biomarker or as therapeutic target in diabetic neuropathy via vitamin K supplementation.

## Supporting information

**S1 Table. Correlations between dp-ucMGP and coronary arterial disease and other microvascular complications of diabetes.**
(DOCX)

## Acknowledgments

We thank Eli Lilly Company, the University of Lausanne, and the clinical staff of the Clinical Investigation Center Paris-Est as well as the Diabetes and Cardiology Departments from the Assistance Publique-Hôpitaux de Paris Pitié-Salpêtrière Hospital in Paris for their participation in this project.

## Author Contributions

**Conceptualization:** Agnès Hartemann, Olivier Bourron.

**Data curation:** Anne-Caroline Jeannin, Joe-Elie Salem, Cees Vemeer, Chloé Amouyal, Olivier Bourron.

**Formal analysis:** Joe-Elie Salem.

**Investigation:** Carole Elodie Aubert, Marine Halbron, Olivier Bourron.

**Methodology:** Olivier Bourron.

**Supervision:** Olivier Bourron.

**Writing – original draft:** Anne-Caroline Jeannin, Olivier Bourron.

**Writing – review & editing:** Joe-Elie Salem, Ziad Massy, Franck Phan, Marine Halbron, Christian Funck-Brentano, Agnès Hartemann, Olivier Bourron.

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
