## [Decision Letter · Decision Letter 0]

23 Dec 2019

PONE-D-19-25013

Inactive Matrix Gla Protein serum levels are associated with peripheral neuropathy in type 2 diabetes

PLOS ONE

Dear Dr. Bourron,

Thank you for submitting your manuscript to PLOS ONE. After careful consideration, we feel that it has merit but does not fully meet PLOS ONE’s publication criteria as it currently stands. Therefore, we invite you to submit a revised version of the manuscript that addresses the points raised during the review process.

As you will recognize from the comments of the reviewer points of critique, especially regarding statistical analysis and presentation of data were raised.

We would appreciate receiving your revised manuscript within 2 months. To enhance the reproducibility of your results, we recommend that if applicable you deposit your laboratory protocols in protocols.io, where a protocol can be assigned its own identifier (DOI) such that it can be cited independently in the future. For instructions see: http://journals.plos.org/plosone/s/submission-guidelines#loc-laboratory-protocols

We look forward to receiving your revised manuscript.

Kind regards,

Rudolf Kirchmair

Academic Editor

PLOS ONE

Journal Requirements:

"This work was supported by a fund from the Lilly Company. The research activities of C.E.A. were supported by a doctoral research scholarship from the University of Lausanne. A.C.J  received a grant from Ministère français des Affaires Sociales et de la Santé (Bourse année recherche 2017/2018)".

We note that you received funding from a commercial source: Lilly Company.

4. Please include your tables as part of your main manuscript and remove the individual files. Please note that supplementary tables (should remain/ be uploaded) as separate "supporting information" files.

Reviewers' comments:

Reviewer's Responses to Questions

**Comments to the Author**

1. Is the manuscript technically sound, and do the data support the conclusions?

Reviewer #1: Yes

2. Has the statistical analysis been performed appropriately and rigorously? 

Reviewer #1: No

3. Have the authors made all data underlying the findings in their manuscript fully available?

Reviewer #1: Yes

4. Is the manuscript presented in an intelligible fashion and written in standard English?

Reviewer #1: Yes

5. Review Comments to the Author

Reviewer #1: The manuscript “Inactive Matrix Gla Protein serum levels are associated with peripheral neuropathy in type 2 diabetes” is well written and provides novel information about the association of inactive form of MGP, dpucMGP and diabetic nephropathy. The authors have properly designed and carried out this study. However, there are some minor issues that should be addressed

1. The title of the manuscript “Inactive Matrix Gla Protein serum levels….”, while in the methodology section the authors state that plasma dpucMGP levels were measured. Which is it?

2. Since dpucMGP has been repeatedly associated with vascular calcification and cardiovascular disease and diabetes complications such as diabetic nephropathy, it would be interesting to show any associations between circulating dpucMGP and coronary arterial disease, retinopathy , eGFR and albuminuria.

3. In the discussion part, the authors state that “Insulin use in Type 2 Diabetes reflects indirectly diabetes duration and poor glycemic control”. Given the fact that BMI includes height and insulin use reflects diabetes duration and HBA1c, how can the authors eliminate the possibility of statistical overlapping in multivariate analysis, where all these variables were included?

4. The discussion part should be enriched with data showing that dpucMGP has been associated with several micro and macro vascular complications of diabetes, including diabetic nephropathy, retinopathy, vascular stiffness and calcification (suggested references to be added: doi:10.3390/ijms20030628, doi: 10.1038/s41598-018-33257-6, doi: 10.1093/ajh/hpy079, doi: 10.1159/000443426 and doi: 10.1016/j.jdiacomp.2017.06.012)

6. PLOS authors have the option to publish the peer review history of their article (what does this mean?). If published, this will include your full peer review and any attached files.

Reviewer #1: No

---

## [Author Response · Author response to Decision Letter 0]

16 Jan 2020

We thank the reviewers for carefully reading our article. As requested we have highlighted changes made from the original version in a separate file labeled 'Revised Manuscript with Track Changes'. An unmarked version of our revised paper without tracked changes was also made. This file was uploaded as separate file and labeled 'Manuscript'.

1. The title of the manuscript “Inactive Matrix Gla Protein serum levels….”, while in the methodology section the authors state that plasma dp-ucMGP levels were measured. Which is it?

Thanks to the reviewer for pointing out this error. We changed the title as follow: « Inactive Matrix Gla Protein serum plasma levels are associated with peripheral neuropathy in Type 2 Diabetes »

2. Since dpucMGP has been repeatedly associated with vascular calcification and cardiovascular disease and diabetes complications such as diabetic nephropathy, it would be interesting to show any associations between circulating dpucMGP and coronary arterial disease, retinopathy , eGFR and albuminuria.

In our cohort, we confirmed that dp-ucMGP was significantly associated with eGFR . 

coronary arterial disease (r; p-value): 0.051; 0.48

Lasered retinopathy (r; p-value): -0.134; 0.06

eGFR (MDRD) (r; p-value): -0.377; < 0.0001

Albuminuria*(r; p-value): -0.268; 0.0001

S1 Table. correlations between dp-ucMGP and coronary arterial disease and other micro-vascular complications of diabetes. r is calculated by Pearson correlation test (dp-ucMGP being normally distributed). Correlations are significant if p<0.05. Significant results are presented in bold. * defined by urinary albumin/creatinine ratio >3 mg/mmol.

We have included, line 273 page 11, in the discussion, a paragraph about these data:

There is some data showing that dp-ucMGP is associated with several micro and macro-vascular complications of diabetes, including diabetic nephropathy, retinopathy, vascular stiffness and vascular calcification(39-41). Although dp-ucMGP has been repeatedly associated with vascular calcification and cardiovascular disease, dp-ucMGP is not associated in our study with coronary arterial disease (S1 table)(39, 41). As observed by others, dp-ucMGP is associated negatively with eGFR estimated by MDRD and with albuminuria (S1 table) (39, 40). However we don’t find any association between dp-ucMGP and retinopathy treated by laser (S1 table) despite some data suggesting that dp-ucMGP could be a marker of retinal health(42). Additional studies are needed to explore specifically these associations in patients with diabetes.

We have added the S1table in supplementary data. 

3. In the discussion part, the authors state that “Insulin use in Type 2 Diabetes reflects indirectly diabetes duration and poor glycemic control”. Given the fact that BMI includes height and insulin use reflects diabetes duration and HBA1c, how can the authors eliminate the possibility of statistical overlapping in multivariate analysis, where all these variables were included?

We agree with the reviewer that this statement in the discussion is misleading and we dropped it.

None of these parameters (BMI, diabetes duration and HbA1C level) were associated with presence of neuropathy in univariate analysis, as compared to insulin use and height which were (see table below). 

Characteristics Total cohort Neuropathy (NDS≥6) Without neuropathy (NDS<6) p-value

BMI (Kg/m2) 29,16±5.3 30,23±5.5 28,97±5.2 ns

Diabetes duration, years 14.6±9.3 14.6±10.2 14.6±9.2 ns

HbA1c, mmol/mol 61.8±16.2 66.6±20.5 60.9±15.2 ns

HbA1c, % 7.8±1.5 8,2±1.9 7.7±1.4 ns

Height (cm) 170±8 173±7 169±8 0.009

Insulin treatment, n(%) 94 (47.5) 24(77.4) 70(41.9) 0.0003

We have modified the paragraph corresponding (line 216 p9) as follow:

We don’t find any significant association between BMI and neuropathy in our study, probably because BMI formula includes height, which is maybe a more important marker of diabetic peripheral neuropathy due to the length-dependent presentation of this neuropathy(20). Furthermore, BMI is not associated with neuropathy in the DIACART study also probably because majority of the patients included were overweight or obese (mean BMI 29.16+/-5.3 Kg/m2). Insulin use in Type 2 Diabetes reflects indirectly diabetes duration and poor glycemic control. So that is probably why it is associated with diabetic neuropathy in the DIACART study. Although insulin is considered as a neurotrophic factor and although low-dose insulin can have beneficial effects on diabetic neuropathy, insulin use is associated with diabetic neuropathy in the DIACART study(21). Retinopathy and nephropathy are usual comorbidities of diabetic neuropathy, explaining their association in this study(4). BMI was not associated with neuropathy in the DIACART study probably because majority of the patients were overweight or obese (mean BMI 29.16+/-5.3 Kg/m2). 

4. The discussion part should be enriched with data showing that dpucMGP has been associated with several micro and macro vascular complications of diabetes, including diabetic nephropathy, retinopathy, vascular stiffness and calcification (suggested references to be added: doi:10.3390/ijms20030628, doi: 10.1038/s41598-018-33257-6, doi: 10.1093/ajh/hpy079, doi: 10.1159/000443426 and doi: 10.1016/j.jdiacomp.2017.06.012)

We have added these references in the discussion.

---

## [Decision Letter · Decision Letter 1]

31 Jan 2020

Inactive Matrix Gla Protein plasma levels are associated with peripheral neuropathy in type 2 diabetes

PONE-D-19-25013R1

Dear Dr. Bourron,

We are pleased to inform you that your manuscript has been judged scientifically suitable for publication and will be formally accepted for publication once it complies with all outstanding technical requirements.

With kind regards,

Rudolf Kirchmair

Academic Editor

PLOS ONE

Additional Editor Comments (optional):

Reviewers' comments:

Reviewer's Responses to Questions

**Comments to the Author**

1. If the authors have adequately addressed your comments raised in a previous round of review and you feel that this manuscript is now acceptable for publication, you may indicate that here to bypass the “Comments to the Author” section, enter your conflict of interest statement in the “Confidential to Editor” section, and submit your "Accept" recommendation.

Reviewer #1: All comments have been addressed

2. Is the manuscript technically sound, and do the data support the conclusions?

Reviewer #1: Yes

3. Has the statistical analysis been performed appropriately and rigorously? 

Reviewer #1: Yes

4. Have the authors made all data underlying the findings in their manuscript fully available?

Reviewer #1: Yes

5. Is the manuscript presented in an intelligible fashion and written in standard English?

Reviewer #1: Yes

6. Review Comments to the Author

Reviewer #1: all comments were adequally answered and the revised version of the manuscript is feasible for publication

7. PLOS authors have the option to publish the peer review history of their article (what does this mean?). If published, this will include your full peer review and any attached files.

Reviewer #1: No

---

## [Editor Report · Acceptance letter]

11 Feb 2020

PONE-D-19-25013R1 

Inactive Matrix Gla Protein plasma levels are associated with peripheral neuropathy in type 2 diabetes 

Dear Dr. Bourron:

I am pleased to inform you that your manuscript has been deemed suitable for publication in PLOS ONE. Congratulations! Your manuscript is now with our production department. 

With kind regards,

on behalf of

Prof Rudolf Kirchmair 

Academic Editor

PLOS ONE